# Threshold of increase in oxygen demand to predict mechanical ventilation use in novel coronavirus disease 2019: A retrospective cohort study incorporating restricted cubic spline regression

Ryo Yamamoto[1]*, Ryo Takemura[2], Asako Yamamoto[2], Kazuki Matsumura[1], Daiki Kaito[1], Koichiro Homma[1], Michihiko Wada[2], Junichi Sasaki[1], on behalf of Keio Donner Project¶

1 Department of Emergency and Critical Care Medicine, Keio University School of Medicine, Tokyo, Japan,
2 Clinical and Translational Research Center, Keio University Hospital, Tokyo, Japan

¶ Membership of the Keio Donner Project is listed in the Acknowledgments.
* ryo.yamamoto@gmail.com

**Data Availability Statement:** The data of this study are available from the Donner Registry of the Keio

## Abstract

### Background

Rapid deterioration of oxygenation occurs in novel coronavirus disease 2019 (COVID-19), and prediction of mechanical ventilation (MV) is needed for allocation of patients to intensive care unit. Since intubation is usually decided based on varying clinical conditions, such as required oxygen changes, we aimed to elucidate thresholds of increase in oxygen demand to predict MV use within 12 h.

### Methods

A single-center retrospective cohort study using data between January 2020 and January 2021was conducted. Data were retrieved from the hospital data warehouse. Adult patients diagnosed with COVID-19 with a positive polymerase chain reaction (PCR) who needed oxygen during admission were included. Hourly increments in oxygen demand were calculated using two consecutive oxygen values. Covariates were selected from measurements at the closest time points of oxygen data. Prediction of MV use within 12 h by required oxygen changes was evaluated with the area under the receiver operating curves (AUCs). A threshold for increased MV use risk was obtained from restricted cubic spline curves.

### Results

Among 66 eligible patients, 1835 oxygen data were analyzed. The AUC was 0.756 for predicting MV by oxygen demand changes, 0.888 by both amounts and changes in oxygen, and 0.933 by the model adjusted with respiratory rate, PCR quantification cycle (Ct), and days from PCR. The threshold of increments of required oxygen was identified as 0.44 L/min/h and the probability of MV use linearly increased afterward. In subgroup analyses, the

Donner Project; however, restrictions apply to the availability of these data, which were used under a license for the current study, so they are not publicly available. However, data are available from the Keio Donner Project at Keio University School of Medicine (contact via email: sayuri.z2@keio.jp) for researchers who meet the criteria for access to confidential data.

**Funding:** The authors received no specific funding for this work.

**Competing interests:** The authors have declared that no competing interests exist.

threshold was lower (0.25 L/min/h) when tachypnea or frequent respiratory distress existed, whereas it was higher (1.00 L/min/h) when viral load is low (Ct $\geq$20 or days from PCR >7 days).

## Conclusions

Hourly changes in oxygen demand predicted MV use within 12 h, with a threshold of 0.44 L/min/h. This threshold was lower with an unstable respiratory condition and higher with a low viral load.

## Introduction

Novel coronavirus disease 2019 (COVID-19) involves lung tissue injury and often causes respiratory failure that requires mechanical ventilation (MV) [1, 2]. As the pandemic of COVID-19 has significantly depleted medical resources worldwide, the allocation of patients to appropriate places, such as the intensive care unit (ICU), general ward, and home, is needed to prevent unfavorable clinical outcomes due to insufficient treatment [3, 4]. However, quick oxygenation deterioration has been reported in patients with COVID-19 compared to other lung diseases. This situation impedes physicians from forecasting the need for MV in advance [5, 6].

While several studies have attempted to develop a prediction model for the need of MV or ICU admission, there is no well-accepted method that captures rapidly changing respiratory status in patients with COVID-19 [7–12]. Although some clinical scoring systems showed promising results with high discrimination, most use daily clinical data or those on admission and only predict deterioration within 24–48 h or thereafter [8, 9, 11, 12]. Since candidates for MV are usually on oxygen therapy and changes in respiratory status are frequently assessed within a day, estimation with such a long-term interval using a score is not practical. Moreover, although machine learning incorporating vital signs, laboratory data, and images could accurately calculate the risks for MV [7, 10] it would be difficult for most health care facilities to adopt the complicated program without trained experts.

Given that the decision to intubate patients with COVID-19 largely depends on oxygenation deterioration [13], an hourly increase in oxygen demand and the amount of required oxygen would be important parameters to determine the need for MV within a short time. Accordingly, we examined the clinical consequences of patients with COVID-19 who required oxygen, using detailed electronic data obtained directly from a hospital information system that recorded various kinds of information related to oxygen therapy. We aimed to elucidate whether an increase in oxygen demand would predict MV use within 12 h, with a hypothesis that an hourly increase of supplemental oxygen higher than a specific threshold would be associated with an increased risk of MV within 12 h.

## Materials and methods

### Study design and setting

We conducted a single-center retrospective cohort study using data between January 2020 and January 2021, that was obtained directly from the hospital information system of Keio University Hospital, a tertiary care center in Tokyo, Japan.

Sporadic COVID-19 cases were noted in Japan in January 2020. The governor of Tokyo Metropolis announced the first stay-at-home order in April 2020, which lasted one month, then the second in January 2021 [14]. There were three surges of newly diagnosed COVID-19 cases during the study period. During these surges, several academic organizations were concerned with nosocomial infection among healthcare providers during the invasive respiratory care of patients with COVID-19 [13, 15–17]; therefore, they recommended avoiding non-invasive positive-pressure ventilation (NIPPV) and high-flow nasal cannula (HFNC) for patients with COVID-19 and intubating patients with a relatively low oxygen flow threshold, such as 6–8 L/min.

At the study institution, patients with mild to moderate COVID-19 who required oxygen but not MV were treated with pulmonary internal medicine physicians in general wards. Intensive care physicians treated those with severe COVID-19 who needed MV or extracorporeal membrane oxygenation in the ICU. Daily discussion between the two services was conducted regarding candidates for MV. The need for MV was decided by discussion considering respiratory status, hemodynamic stability, and the oxygen demand mentioned above. Urgent transfer of patients to the ICU due to an unexpected rapid increase in oxygen demand was conducted on a 24-hour basis depending on the agreement of the two services. Patients with severe comorbidity, such as congestive heart failure requiring oxygen and acute kidney injury requiring hemodialysis, were admitted to the ICU regardless of the severity of COVID-19.

## Ethical statement

This study was approved by the Institutional Review Board of the Keio University School of Medicine (application number: 20200063) for conducting research with humans. The requirement for informed consent was waived because of the anonymous nature of the data used.

## Study population

We included patients (1) aged ≥20 years, (2) diagnosed as COVID-19 with a positive reverse transcription polymerase chain reaction (RT-PCR) result for severe acute respiratory syndrome coronavirus-2 (SARS-CoV-2) from an upper respiratory tract sample obtained by nasopharyngeal swab, and (3) on oxygen therapy at any time during admission. Patients who were intubated on the day of admission and those with unknown or missing data on the amount of oxygen administered were excluded. Patients who were intubated only for airway management in scheduled surgery were also excluded.

All recorded data of the amount of oxygen administered were examined individually, even in the same patient. However, data on the amount of oxygen administered after MV was initiated were not included in this study.

## Data collection and definition

Data were obtained from the Donner Registry, established as a real-world data registry by the Keio Donner Project, a COVID-19 research group at Keio University School of Medicine. The Donner Registry has been prospectively collecting data of patients with COVID-19 from the hospital information system with every record related to patient care. In the hospital information system, several record types, such as demographic data, auto-recorded parameters in patient-monitoring devices, descriptive records by health care providers, laboratories, images, and detailed information of when these data were saved, are archived in different systems. The Donner Registry has collected data using a data warehouse connected to all records in the hospital information system. This registry is maintained by designated data managers of the Keio

Donner Project. Patient data related to this study were also obtained by the data manager, who was blinded to study analyses.

Collected data included patient demographics; comorbidities, such as chronic obstructive pulmonary disease (COPD), interstitial pneumonia (IP), asthma, congestive heart failure (CHF), chronic kidney disease (CKD), cirrhosis, hypertension, and diabetes mellitus; date of a positive RT-PCR result for SARS-CoV-2; the RT-PCR quantification cycle (Ct) for SARS-CoV-2; vital signs recorded by patient-monitoring devices and health care providers; laboratory data, such as C-reactive protein (CRP), D-dimer, and glucose; medications for COVID-19 with the date of administration, including corticosteroids, remdesivir, tocilizumab, and unfractionated and low-molecular-weight heparin; the amount of oxygen administered (L/min); a descriptive record of the existence of respiratory distress; and the time (h and min) for each collected data. The time when intubation was performed, hospital length of stay, ICU length of stay, days of MV use, and survival status were also available.

Change in oxygen demand was defined as a change in the amount of administered oxygen per hour, calculated using two consecutive oxygen data. Vital signs and blood glucose associated with oxygen data were determined as those measured at the closest time points prior to the oxygen data; those measured >24 h prior to the oxygen data were not used. Similarly, laboratory data associated with oxygen data were determined as those measured within three days before the oxygen data. Respiratory distress was defined as distress symptoms recorded at any time in 6-hour periods. The frequency of respiratory distress in a day was shown with a 0–4 scale, defined as the number of respiratory distress events during the past 24 h (four 6-hour periods).

## Outcome measures

The primary outcome was the initiation of MV within 12 h, defined as intubation conducted within 12 h after the time point of oxygen data. Secondary outcomes included 90-day mortality and ICU- and ventilator-free days up to day 30, in which the days were counted from the day of each oxygen data.

## Statistical analysis

A receiver operating curve (ROC) was used to determine the ability to predict MV use by changes in oxygen demand. Then, the area under the ROC (AUC) was compared with several adjusted models to evaluate the clinical usefulness of changes in oxygen demand. Relevant covariates were carefully selected from known or possible predictors for deteriorating oxygenation based on previous studies [18–22], including age, body mass index, comorbidities (COPD, IP, asthma, CHF, CKD, and cirrhosis), days from diagnosis of COVID-19, Ct value of initial RT-PCR for SARS-CoV-2, days from the initiation of medications for COVID-19 (corticosteroids, remdesivir, and tocilizumab), vital signs (respiratory rate [RR], heart rate, and systolic blood pressure [SBP]), laboratories (CRP, D-dimer, and glucose), and the frequency of respiratory distress in a day (0–4 scale). Adjusted models were developed using multivariate logistic regression analyses, in which variables were entered using the stepwise or simultaneous method. Variables for the full adjusted model were selected based on a point estimate for odds ratio or degree of α error. The number of selected variables were limited to avoid over-fitting; 5–10 outcomes for each potential predictor.

The clinical usefulness of increased oxygen demand to predict MV use was assessed in unadjusted and adjusted models, using sensitivity, specificity, negative predictive value (NPV), and positive predictive values (PPV). Moreover, the restricted cubic spline regression model was used to identify the threshold for rapidly increasing risks for MV within 12 h [23]. The

spline curve was drawn to show the risks for MV use by oxygen demand increases, then an inflection point of the spline curve was determined as the threshold, considering an increase of absolute risk from the baseline of >1%.

Sensitivity analysis was conducted by excluding negative changes in oxygen demand. Furthermore, the association between secondary outcomes and the changes in oxygen demand was also analyzed using logistic and linear regression models.

Subgroup analysis was performed to examine the relationship between changes in oxygen demand, clinical characteristics, and the requirement of MV. Calculating AUC and identification of the threshold of increment in oxygen based on spline curves were repeated in the subgroup of patients who were divided based on age (<65 vs. ≥65 years), the amount of administered oxygen (<4 vs. ≥4 L/min), RR (<20 vs. ≥20 /min), days from diagnosis of COVID-19 with RT-PCR (≤7 vs. >7 days), degree of viral load (Ct value of initial RT-PCR <20 vs. ≥20), and frequency of respiratory distress in a day (<2 vs. ≥2 in 0–4 scale).

Descriptive statistics are presented as the median (interquartile range [IQR]) or a number (percentage). Results are shown using standardized differences and the 95% confidence interval (CI). In hypothesis testing, a two-sided α threshold of 0.05 was considered statistically significant. Considering the low number of included data points, optimism was evaluated with bootstrapping (resampling the model 1000 times) to obtain a corrected AUC [24]. All statistical analyses were conducted using SAS version 9.4 (SAS Institute Inc., Cary, NC).

## Results

### Patient characteristics

Among 285 patients with COVID-19 during the study period, 72 adults had oxygen therapy and met all the inclusion criteria. A total of 6 patients were intubated on the day of admission; therefore, 66 patients were eligible for this study. Among 2524 oxygen data available in included patients, 689 were excluded from the analyses because they were after the MV initiation. The patient flow diagram is shown in Fig 1.

Patient characteristics are shown in Table 1. Eleven patients (16.7%) required MV during admission and used MV. Patients treated with MV were older and had lower Ct values (higher viral load) on RT-PCR for SARS-CoV-2 (18 vs. 24) than those who did not require MV. Among 1835 oxygen data analyzed in this study, MV was initiated within 12 h after 34 (1.9%) oxygen data. Clinical information associated with each oxygen data is summarized in Table 2. When MV was initiated in the next 12 h, patients had a higher increase in oxygen demand (0.25 vs. 0.00 L/min/h) and higher RR, SBP, D-dimer, and glucose than when MV was not needed in the next 12 h. Days from the positive PCR test, admission, and initiation of corticosteroid, remdesivir, and unfractionated heparin were fewer when MV was required in the next 12 h, compared with when it was not required (4 vs. 7 days from PCR, 5 vs. 11 days from admission, 1 vs. 6 days from corticosteroids, 1 vs. 5 days from remdesivir, and 0 vs. 6 days from unfractionated heparin, respectively). Conversely, the frequency of respiratory distress was comparable regardless of the need for MV in the next 12 h.

**Prediction of MV use and secondary outcomes.** Accuracy in predicting the need of MV within 12 h by the changes in oxygen demand was assessed in several logistic regression models. In these analyses, the full adjusted model included RR, Ct value of PCR for SARS-CoV-2, and days from positive PCR as covariates. AUC was 0.756 (95% CI, 0.662–0.851) in the simple model using only changes in oxygen demand, 0.888 (0.856–0.919) in a combination model using both amounts and changes in oxygen demand, and 0.933 (0.908–0.958) in the full adjusted model (Fig 2 and Table 3). All models had >99% of NPV at the Youden index, whereas PPV was only 7%–14%. Sensitivity analyses found similar results, in which negative

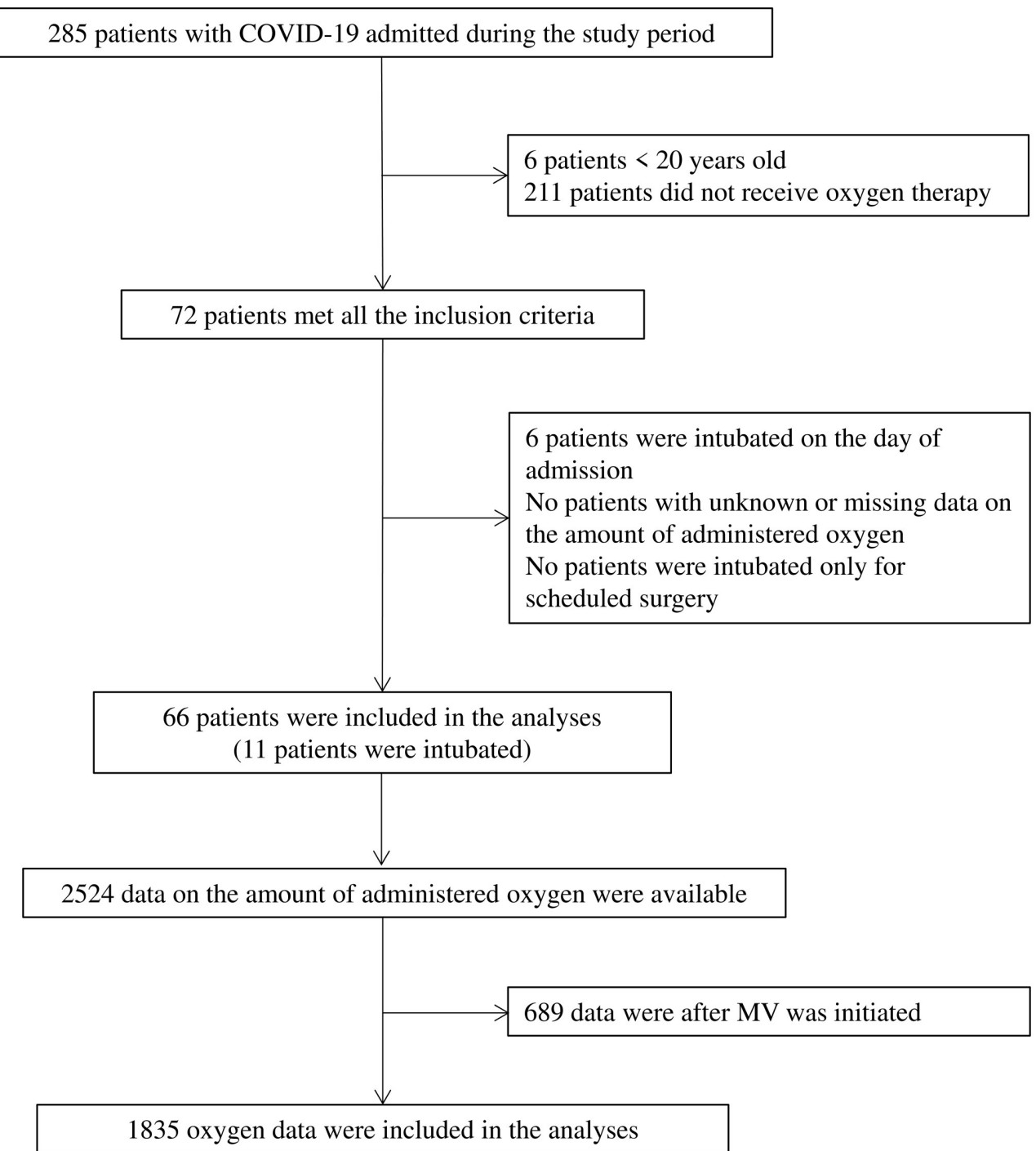

**Fig 1. Patient flow diagram.** Among 285 patients with COVID-19 during the study period, 72 adult patients had oxygen therapy and met all the inclusion criteria. A total of 6 patients were intubated on the day of admission, and therefore 66 patients were eligible for this study. Among 2524 oxygen data available in included patients, 689 were excluded from the analyses because they were those after MV initiation. Abbreviations: COVID-19, novel coronavirus disease 2019; MV, mechanical ventilation.

**Table 1. Characteristics of patients with COVID-19 on oxygen therapy.**

| Characteristics | Intubation | No intubation | p value | Standardized Difference |
|---|---|---|---|---|
| Case | 11 | 55 | | |
| Age, years, median (IQR) | 75 (67–81) | 64 (54–76) | 0.044 | 0.773 |
| Sex, male, n (%) | 10 (90.9%) | 39 (70.9%) | 0.264 | 0.526 |
| BMI, median (IQR) | 25 (22–26) | 26 (22–29) | 0.265 | 0.378 |
| Comorbidity, n (%) | 4 (36.4%) | 19 (34.5%) | 1.000 | 0.038 |
| COPD | 0 (0.0%) | 2 (3.6%) | | |
| Interstitial pneumonia | 0 (0.0%) | 2 (3.6%) | | |
| Asthma | 0 (0.0%) | 0 (0.0%) | | |
| CHF | 0 (0.0%) | 2 (3.6%) | | |
| CKD | 0 (0.0%) | 0 (0.0%) | | |
| Cirrhosis | 0 (0.0%) | 0 (0.0%) | | |
| Hypertension | 4 (36.4%) | 11 (20.0%) | | |
| Diabetes mellitus | 0 (0.0%) | 6 (10.9%) | | |
| Smoking history, n (%) | 2 (18.2%) | 10 (18.2%) | 1.000 | 0.009 |
| Ct value on RCP for SARS-CoV-2[a], median (IQR) | 18 (14–24) | 24 (20–31) | 0.002 | 1.042 |
| Treatment, n (%) | | | | |
| Corticosteroid | 6 (54.5%) | 22 (40.0%) | 0.507 | 0.294 |
| Tocilizumab | 0 (0.0%) | 8 (14.5%) | 0.334 | 0.582 |
| Remdesivir | 5 (45.5%) | 20 (36.4%) | 0.735 | 0.186 |
| Unfractionated heparin | 8 (72.7%) | 25 (45.5%) | 0.185 | 0.577 |

COVID-19 = Novel coronavirus disease 2019, IQR = interquartile range, COPD = chronic obstructive pulmonary disease, CHF = congestive heart failure,

CKD = chronic kidney disease, Ct = cycle of quantification, PCR = polymerase chain reaction, and SARS-CoV-2 = severe acute respiratory syndrome coronavirus 2.

[a]When multiple samples were obtained at the same time, Ct values were averaged.

**Table 2. Clinical information associated with changes in oxygen demand.**

| | Intubation within 12 h | No intubation within 12h | p-value | Standardized Difference |
|---|---|---|---|---|
| Number of data points | 34 | 1801 | | |
| Changes in oxygen demand, L/min/h, median (IQR) | 0.25 (0.00–0.67) | 0.00 (0.00–0.00) | <0.001 | 0.709 |
| Vital signs, median (IQR) | | | | |
| Respiratory rate, /min | 21 (16–26) | 18 (16–20) | 0.003 | 0.500 |
| Heart rate, /min | 72 (65–78) | 75 (64–86) | 0.434 | 0.145 |
| SBP, mmHg | 130 (126–134) | 118 (106–130) | <0.001 | 0.766 |
| Days from positive PCR, median (IQR) | 4 (1–9) | 7 (2–16) | 0.002 | 0.713 |
| Days from admission, median (IQR) | 5 (1–15) | 11 (5–45) | 0.005 | 0.534 |
| Duration of treatment, days from, median (IQR) | | | | |
| Corticosteroid | 1 (0–5) | 6 (5–20) | 0.009 | 0.826 |
| Tocilizumab | N/A | 3 (0–5) | N/A | N/A |
| Remdesivir | 1 (0–5) | 5 (1–14) | 0.025 | 0.812 |
| Unfractionated heparin | 0 (0–2) | 6 (1–14) | <0.001 | 1.070 |
| Laboratory, median (IQR) | | | | |
| CRP | 1.4 (1.0–1.7) | 2.5 (0.9–8.2) | 0.106 | 0.192 |
| D-dimer | 9.2 (4.9–18.8) | 3.6 (1.5–8.1) | <0.001 | 0.885 |
| Glucose | 148 (99–166) | 176 (136–238) | 0.014 | 0.481 |
| Frequency of respiratory distress[a], median (IQR) | 1 (0–2) | 1 (0–2) | 0.108 | 0.289 |

IQR = interquartile range, SBP = systolic blood pressure, PCR = polymerase chain reaction, and CRP = C-reactive protein.

[a]Frequency of respiratory distress were shown using 0–4 scale.

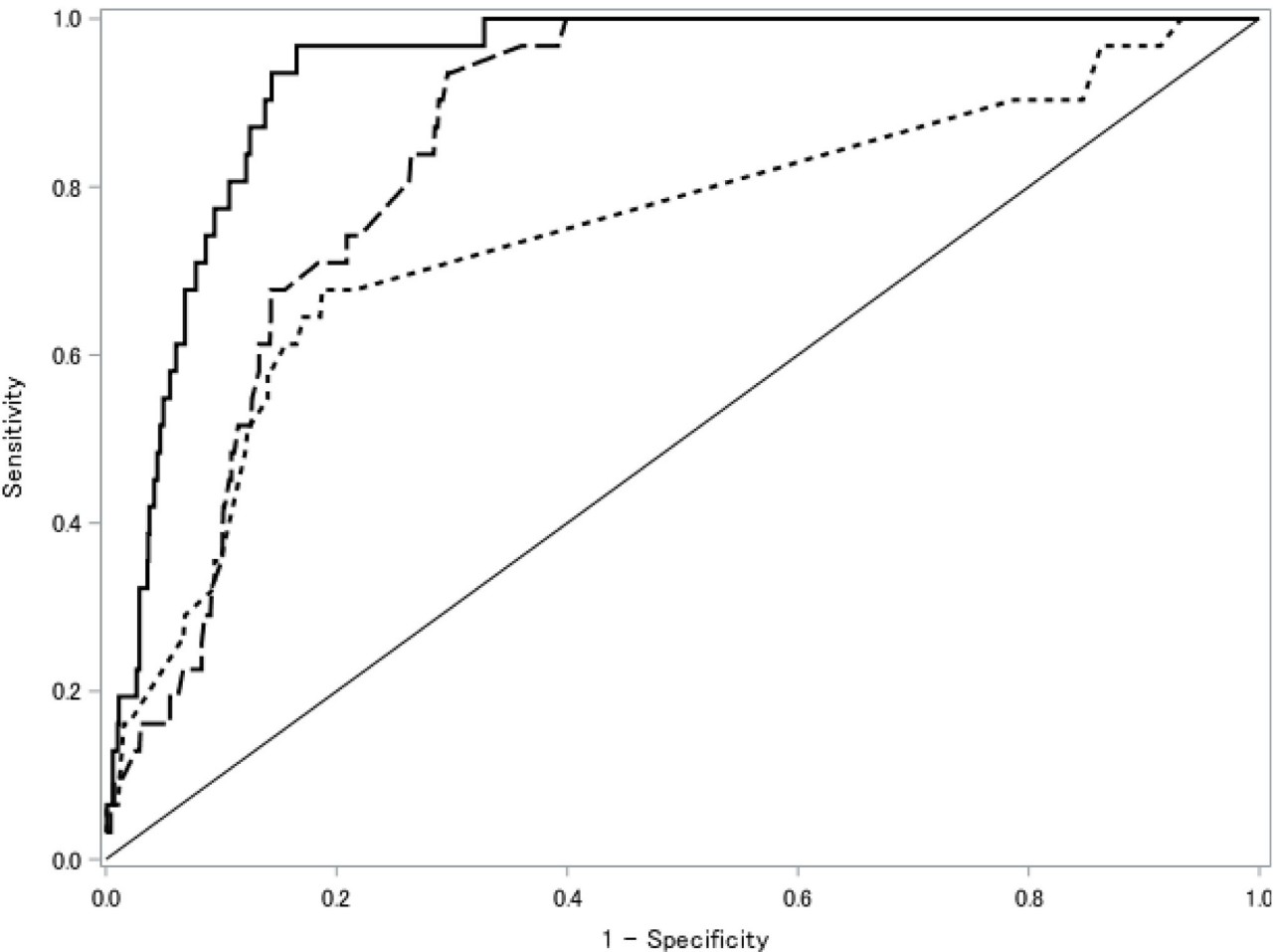

**Fig 2. Receiver operating curve for prediction of mechanical ventilation use by changes in oxygen demand.** Changes in oxygen demand to predict MV use within 12 h were evaluated by ROCs in several models as follows: simple model only using increments in oxygen demand (AUC 0.756 [0.662–0.851]); combination model using both amounts and increments in oxygen demand (AUC 0.888 [0.856–0.919]); and a fully adjusted model including amounts and increments in oxygen demand, RR, Ct value of PCR for SARS-CoV-2, and days from positive PCR (0.933 [0.908–0.958]). Abbreviations: ROC, Receiver operating curve; MV, mechanical ventilation; AUC, area under the ROC; RR, respiratory rate; Ct, quantification cycle; PCR, polymerase chain reaction; and SARS-CoV-2, severe acute respiratory syndrome coronavirus 2.

changes in oxygen demand were excluded (S1 Fig and Table 3). Optimism was evaluated using bootstrapping in each model, which identified corrected AUCs similar to original AUCs.

A restricted cubic spline curve was drawn in Fig 3. Based on the inflection point in the spline curve, a 0.44 L/min/h increase in oxygen demand was identified as the threshold to predict MV in the next 12 h, where the NPV was 98.5%. With a higher oxygen increase than the threshold, the probability of MV use linearly increased.

Analyses on secondary outcomes revealed that increments in oxygen demand were associated with increased 90-day mortality but not with ICU- and ventilator-free days that were counted from the day of each oxygen data (S1 Table).

**Subgroup analyses.** In the subgroup analyses (Table 4), the high accuracy in predicting the requirement of MV by the increments in oxygen demand was observed: >0.9 of AUCs and >98% of NPV at the thresholds were found in most subgroups (Table 4).

The threshold for increased risk of MV use was lower in a patient with a RR ≥20/min than those with a RR <20/min, as well as days from positive PCR ≤7 than >7, high viral load

**Table 3. Accuracy for prediction of MV usage by changes in oxygen demand.**

| Model # | Variables in model | AUC | 95% CI | Optimism | Corrected AUC | Sensitivity | Specificity | NPV | PPV |
|---|---|---|---|---|---|---|---|---|---|
| 1 | Changes in oxygen demand | 0.756 | 0.662–0.851 | 0.001 | 0.756 | 69.7% | 81.9% | 99.3% | 6.8% |
| 2 | Amounts and changes of oxygen demand | 0.888 | 0.856–0.919 | 0.002 | 0.885 | 87.9% | 79.6% | 99.7% | 7.5% |
| 3 | Amounts and changes of oxygen demand with other predictors[a] | 0.933 | 0.908–0.958 | 0.005 | 0.924 | 96.8% | 83.5% | 99.9% | 13.5% |
| 4 | Increments in oxygen demand[b] | 0.774 | 0.689–0.860 | 0.001 | 0.775 | 76.7% | 77.2% | 99.3% | 6.8% |
| 5 | Amounts and increments of oxygen demand | 0.873 | 0.834–0.911 | 0.002 | 0.870 | 86.7% | 76.4% | 99.6% | 7.4% |
| 6 | Amounts and increments of oxygen demand with other predictors[a] | 0.927 | 0.897–0.957 | 0.007 | 0.920 | 96.4% | 82.5% | 99.9% | 14.4% |

MV = mechanical ventilation, AUC = area under the receiver operating characteristic curve, CI = confidence interval, NPV = negative predictive value, and

PVV = positive predictive value. Sensitivity, specificity, NPV, and PPV were calculated with Youden Index.

[a]Other predictors included Ct value of polymerase chain reaction (PCR), days from positive PCR, and respiratory rate (RR).

[b]Analyses were performed after excluding negative changes in oxygen demand.

Logit-transformed predictive rate for MV usage within 12 h was calculated in each model as follows:

(1) $0.97 \times$ changes in oxygen $- 4.19$

(2) $0.56 \times$ changes in oxygen $+ 0.21 \times$ amounts of oxygen $- 4.98$

(3) $0.54 \times$ changes in oxygen $+ 0.21 \times$ amounts of oxygen $- 0.12 \times$ Ct value $- 0.12 \times$ days from PCR $+ 0.02 \times$ RR $- 1.70$

(4) $0.95 \times$ changes in oxygen $- 4.15$

(5) $0.55 \times$ changes in oxygen $+ 0.20 \times$ amounts of oxygen $- 4.87$

(6) $0.52 \times$ changes in oxygen $+ 0.20 \times$ amounts of oxygen $- 0.12 \times$ Ct value $- 0.14 \times$ days from PCR $+ 0.01 \times$ RR $- 1.47$

(Ct < 20) than low viral load (Ct ≥ 20), and high frequency of respiratory distress (≥2 in 0–4 scale) than low frequency (<2 in 0–4 scale). Conversely, thresholds were similar regardless of the amount of administered oxygen (0.33 L/min/h in low amount oxygen use [<4 L/min] vs. 0.40 L/min/h in high amount oxygen use [≥4 L/min]).

## Discussion

In this retrospective study, hourly changes in oxygen demand had a high discrimination power to predict MV use, particularly when incorporated with the amount of oxygen, RR, Ct value of PCR, and days from positive PCR. Notably, an increment in oxygen demand higher than 0.44 L/min/h significantly increased the risk for the requirement of MV in the next 12 h.

Several reasons would be considered behind the high predictive ability for the need for MV in this study. First, hourly changes in oxygen demand would be a highly reliable predictor of MV use because most physicians intubate patients when oxygen demand increases, particularly with an accelerated increase [25]. Second, the current study analyzed all data related to oxygen therapy at any given time point, which would have captured rapidly changing respiratory status of COVID-19 [26]. While preexisting scores, such as Respiratory Rate Oxygenation and National Early Warning Score, utilized clinical parameters only at defined time points, including on admission and/or a few days after admission [9, 10], each patient in this study had detailed data with nearly 30 different time points. Third, several clinically valuable covariates were also obtained directly from the hospital information system and analyzed along with the changes in oxygen demand. Given that auto-recorded vital signs, days from positive PCR or medications, and frequency of respiratory distress are important information for physicians

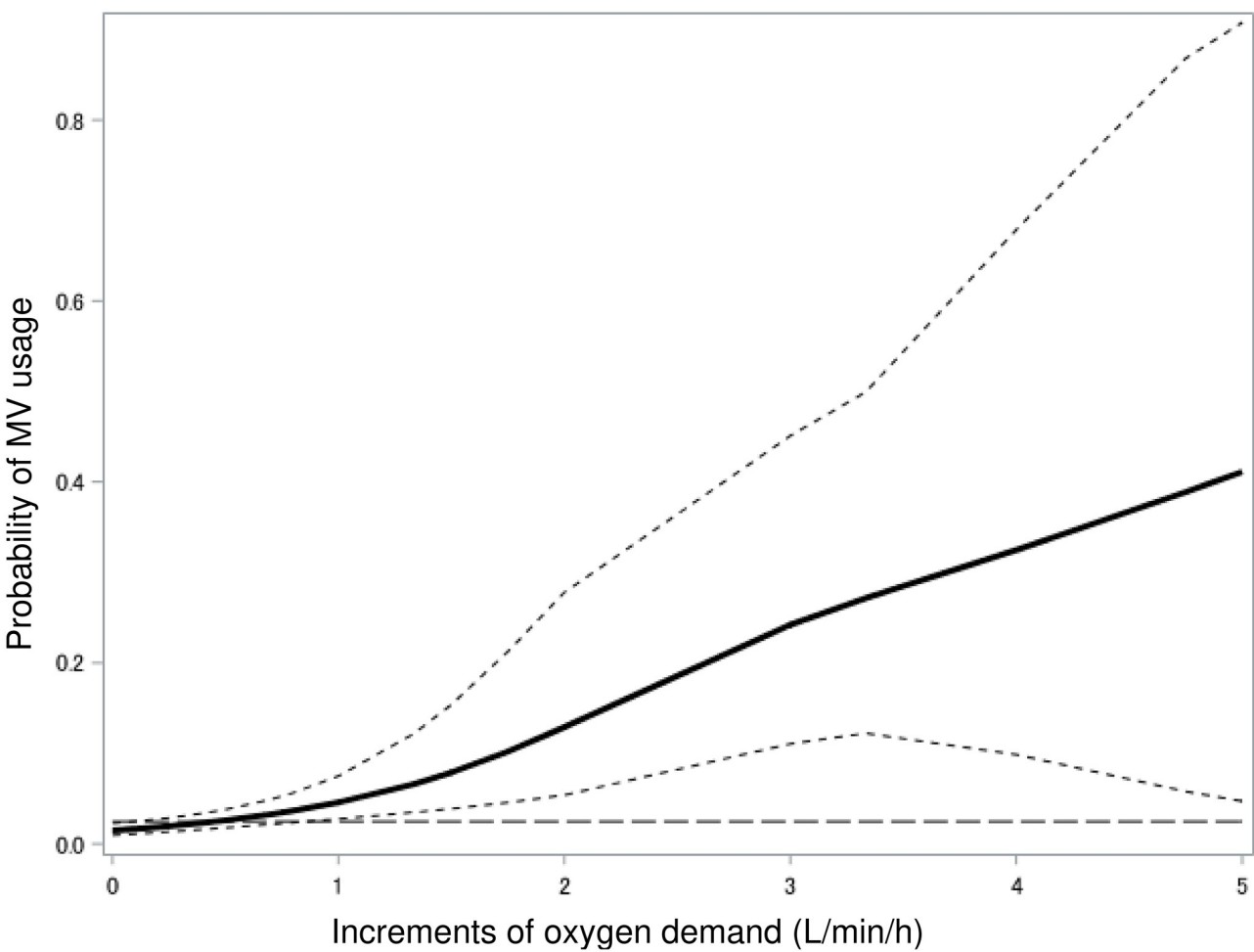

**Fig 3. Restricted cubic spline curves and threshold of increments in oxygen demand.** The restricted cubic spline curve was shown for the risks of MV use within 12 h by increments in oxygen demand, with dashed lines for 95% CI. Based on the inflection point, which considers an increase of absolute risk from the baseline by >1% (horizontal dashed line), 0.44 L/min/h of increment in oxygen demand was identified as the threshold to predict mechanical ventilation in the next 12 h. With a higher increment of oxygen than the threshold, the probability of MV use linearly increased. Abbreviations: MV, mechanical ventilation; CI, confidence interval.

to decide intubation [2, 19, 26, 27], utilizing such variables with oxygen data would result in high predictive power.

According to the current results, using the increments in oxygen demand to forecast the need for MV has various merits. As the prediction window was 12 h in this study, alternation of administered oxygen in the daytime would help physicians determine to transfer a patient to the ICU before the night. In addition, considering that a greater influence on the prediction of MV use was observed in the changes in oxygen demand than the amount of oxygen, an increment in the dose of oxygen would be useful even when a high amount of oxygen is administered. Moreover, NPV for the initiation of MV is as high as >99.5% even in the simple model only utilizing changes in oxygen demand; therefore, the possibility of intubation within 12 h would be denied solely by the lower increment of oxygen than the threshold.

To clinically adopt the threshold of increments in oxygen demand, patient characteristics should be considered because various thresholds for increasing risks for MV use were obtained in subgroup analyses. As the thresholds were lower (0.25 L/min/h) among patients with

**Table 4. Prediction of MV use by increments in oxygen demand in subgroups.**

| | AUC | 95% CI | Threshold (L/min/h)[a] | Sensitivity | Specificity | NPV | PPV |
|---|---|---|---|---|---|---|---|
| Age | | | | | | | |
| <65 years | 0.990 | 0.978–1.000 | 0.33 | 66.7% | 88.9% | 99.4% | 8.9% |
| > = 65 years | 0.940 | 0.915–0.966 | 0.50 | 33.3% | 88.4% | 98.2% | 6.3% |
| Amount of oxygen | | | | | | | |
| <4 L/min | 0.990 | 0.978–1.000 | 0.33 | 66.7% | 91.9% | 99.9% | 2.4% |
| > = 4L/min | 0.869 | 0.821–0.918 | 0.40 | 40.7% | 75.6% | 94.9% | 10.3% |
| Respiratory Rate | | | | | | | |
| <20 /min | 0.963 | 0.943–0.983 | 0.75 | 30.8% | 92.9% | 99.1% | 5.1% |
| > = 20 /min | 0.887 | 0.832–0.942 | 0.25 | 64.7% | 81.6% | 97.9% | 14.9% |
| Days from positive PCR | | | | | | | |
| < = 7 days | 0.917 | 0.875–0.958 | 0.33 | 47.8% | 86.4% | 98.2% | 9.7% |
| > 7 days | 0.980 | 0.961–0.999 | 1.00 | 42.6% | 92.6% | 99.3% | 6.0% |
| Viral load | | | | | | | |
| Ct <20 | 0.904 | 0.855–0.954 | 0.22 | 68.4% | 81.4% | 98.2% | 14.6% |
| Ct > = 20 | 0.936 | 0.887–0.984 | 1.00 | 18.2% | 94.2% | 99.0% | 3.4% |
| Frequency of respiratory distress (0–4 scale) | | | | | | | |
| < 2 | 0.918 | 0.877–0.959 | 0.67 | 29.4% | 92.3% | 98.7% | 6.3% |
| > = 2 | 0.953 | 0.908–0.998 | 0.25 | 61.5% | 87.0% | 98.6% | 13.1% |

MV = mechanical ventilation, AUC = area under the receiver operating characteristic curve, CI = confidence interval, NPV = negative predictive value, PVV = positive predictive value, PCR = polymerase chain reaction, and Ct = cycle of quantification.

[a]Threshold was obtained from an infection point in the spline curve in each subgroup.

tachypnea (RR ≥20) and high frequency of respiratory distress (≥2 in 0–4 scale), patients with an unstable respiratory condition would need intubation even with low increments in oxygen demand. However, it should be noted that the difference in the amount of oxygen did not affect the threshold of increments in oxygen demand. Moreover, given that thresholds as high as 1.00 L/min/h were observed in patients with low viral load (Ct ≥20) and a considerable duration passed after the positive PCR (>7 days), such a population can stay at general wards even when oxygen demand is increasing, such as a gradual increase by 3–4 L/min in a day time.

The results in this study must be interpreted within the context of the study design. During the study period, NIPPV and HFNC were not used in patients with COVID-19. Therefore, the thresholds to predict the need for NIPPV or HFNC would be different [28, 29], although intolerance of simple oxygen administration through face masks would be highly predicted by the increments in oxygen demand. Another limitation is that the study was conducted at a single center with limited sample size. Although the changes in oxygen demand would influence more the prediction of the requirement of MV than the amounts of oxygen, thresholds obtained in this study should be validated in future studies with large sample sizes. Moreover, in the pandemic of COVID-19, several novel medications have been developed and reported to improve outcomes. Considering that days from the initiation of medications for COVID-19 were fewer when MV was used in the next 12 h than when MV was not used, some medications would affect the relationship between increments of oxygen demand and prediction of intubation. Finally, as we investigated only patients with COVID-19, our results cannot be generalized to potential candidates for MV who need oxygen due to other diseases.

## Conclusions

The hourly changes in oxygen demand highly predict the need for MV in the next 12 h, particularly when incorporated with the amount of oxygen, RR, Ct value of PCR, and days from positive PCR. While the threshold for increasing risks for MV use was determined as 0.44 L/min/h, a lower threshold was observed in patients with an unstable respiratory condition, such as high RR and high frequency of respiratory distress. Patients with low viral load or >7 days after the positive PCR would tolerate considerable increments of oxygen demand. The threshold identified in this study would be useful for appropriately allocating patients to ICU in regions where resources are overwhelmed due to pandemic of COVID-19, while the generalizability of threshold should be validated by a multi-center trial with large sample size.

## Supporting information

**S1 Fig. Receiver operating curve for sensitivity analysis.** Sensitivity analysis was conducted by excluding negative changes in oxygen demand. Increments in oxygen demand to predict mechanical ventilation use within 12 h was evaluated by ROCs in several models as follows: simple model only using increments in oxygen demand (AUC 0.774 [0.689–0.860]); combination model using both amounts and increments in oxygen demand (AUC 0.873 [0.834–0.911]); and a fully adjusted model including amounts and increments in oxygen demand, RR, Ct value of PCR for SARS-CoV-2, and days from positive PCR (0.927 [0.897–0.957]). Abbreviations: ROC, Receiver operating curve; AUC, area under the ROC; RR, respiratory rate; Ct, quantification cycle; PCR, polymerase chain reaction; and SARS-CoV-2, severe acute respiratory syndrome coronavirus 2.
(TIFF)

**S1 Table. Changes in oxygen demand and secondary outcomes.**
(DOCX)

## Acknowledgments

I would like to extend my deepest gratitude to the member of Keio Donner Project; Masayuki Amagai, Hideyuki Saya, and Hiroshi Nishihara.

## Author Contributions

**Conceptualization:** Ryo Yamamoto, Ryo Takemura, Daiki Kaito, Koichiro Homma, Michihiko Wada.

**Data curation:** Ryo Yamamoto, Ryo Takemura, Asako Yamamoto, Michihiko Wada.

**Formal analysis:** Ryo Yamamoto, Ryo Takemura, Asako Yamamoto.

**Investigation:** Ryo Yamamoto.

**Methodology:** Ryo Yamamoto, Michihiko Wada.

**Project administration:** Asako Yamamoto.

**Supervision:** Koichiro Homma, Junichi Sasaki.

**Validation:** Kazuki Matsumura, Daiki Kaito, Koichiro Homma, Junichi Sasaki.

**Writing – original draft:** Ryo Yamamoto.

**Writing – review & editing:** Ryo Yamamoto, Ryo Takemura, Asako Yamamoto, Kazuki Matsumura, Daiki Kaito, Koichiro Homma, Michihiko Wada, Junichi Sasaki.

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
