## [Decision Letter · Decision Letter 0]

29 Apr 2022

PONE-D-22-06778Threshold of Increase in Oxygen Demand to Predict Mechanical Ventilation Use in Novel Coronavirus Disease 2019: A Retrospective Cohort Study Incorporating Restricted Cubic Spline RegressionPLOS ONE

Dear Dr. YAMAMOTO,

Thank you for submitting your manuscript to PLOS ONE. After careful consideration, we feel that it has merit but does not fully meet PLOS ONE’s publication criteria as it currently stands. Therefore, we invite you to submit a revised version of the manuscript that addresses the points raised during the review process.

We look forward to receiving your revised manuscript.

Kind regards,

Muhammad Tarek Abdel Ghafar, M.D

Academic Editor

PLOS ONE

Journal Requirements:

3. One of the noted authors is a group or consortium [Keio Donner Project]. In addition to naming the author group, please list the individual authors and affiliations within this group in the acknowledgments section of your manuscript. Please also indicate clearly a lead author for this group along with a contact email address.

Reviewers' comments:

Reviewer's Responses to Questions

**Comments to the Author**

1. Is the manuscript technically sound, and do the data support the conclusions?

Reviewer #1: Yes

Reviewer #2: Yes

Reviewer #3: Yes

Reviewer #4: Yes

2. Has the statistical analysis been performed appropriately and rigorously? 

Reviewer #1: Yes

Reviewer #2: I Don't Know

Reviewer #3: Yes

Reviewer #4: Yes

3. Have the authors made all data underlying the findings in their manuscript fully available?

Reviewer #1: No

Reviewer #2: Yes

Reviewer #3: Yes

Reviewer #4: Yes

4. Is the manuscript presented in an intelligible fashion and written in standard English?

Reviewer #1: Yes

Reviewer #2: Yes

Reviewer #3: Yes

Reviewer #4: Yes

5. Review Comments to the Author

Reviewer #1: Appreciating your work, I would like to forwards the following minor comments and suggestions:

1. Please include a separate ethical statement in your methods section, that clearly highlights the consent measures used, and the ethical procedures followed.

2. In your conclusion segment, I believe it will give your findings more impact if you can phrase them in terms of public and global health impact.

Reviewer #2: SARS-COV-2 is primarily a respiratory pathogen, hospitalized patients with covid-19 who decompensate need assistant ventilation. Many patients requiring hospitalization have caused a great strain on hospital resources. A simple tool that can effectively predict the potential need of mechanical ventilation would ensure better use of existing resources. The paper by R Yamamoto et al presented a well-written article addressing a very practical question – the prediction of mechanical ventilation among patients with the covid-19 disease. The authors aimed to determine thresholds of increase in oxygen demand to predict mechanical ventilation use within 12 h. The title and abstract are appropriate for the content of the text. The article is well constructed. The main strength of this paper is that it addresses a timely question and provides a promising result for a solution. At the same time, the limited study sample (66 patients included// 11 patients intubated) and relatively early intubation decision (oxygen flow requirements 6-8L) is a problem in terms of extracting wide conclusions from the work.

Some questions the authors might consider:

- The list of comorbidities does not include hypertension or diabetes. These conditions may place a patient with covid-19 at a higher risk of severe illness.

- The definition of “severe comorbidity” (row 128-129) was not clear. Are these patients included in the study?

- Low molecular weight heparin (LMWH) is not included in the list of administrated medications. Patients with covid 19 may trigger a coagulopathy state and affect changes in oxygen demand.

Reviewer #3: Thanks for giving me the chance to review the above mentioned article. I enjoyed reading this new concept and hoped that it helped you care for your patients during the crisis. Definitely, you need more numbers to empower the results and more than one center for sake of generalizability.

Reviewer #4: Thanks to authors.

I have completed the evaluation of the article entitled "Threshold of Increase in Oxygen Demand to Predict Mechanical Ventilation Use in Novel Coronavirus Disease 2019: A Retrospective Cohort Study Incorporating Restricted Cubic Spline Regression”. I have not observed that grammatical errors in this study. It is a nice study written on determining the oxygen need in COVID-19.

Best regards

6. PLOS authors have the option to publish the peer review history of their article (what does this mean?). If published, this will include your full peer review and any attached files.

Reviewer #1: No

Reviewer #2: No

Reviewer #3: No

Reviewer #4: No

---

## [Author Response · Author response to Decision Letter 0]

9 May 2022

A point-by-point response letter is attached to the manuscript.

---

## [Decision Letter · Decision Letter 1]

30 May 2022

Threshold of Increase in Oxygen Demand to Predict Mechanical Ventilation Use in Novel Coronavirus Disease 2019: A Retrospective Cohort Study Incorporating Restricted Cubic Spline Regression

PONE-D-22-06778R1

Dear Dr. YAMAMOTO,

We’re pleased to inform you that your manuscript has been judged scientifically suitable for publication and will be formally accepted for publication once it meets all outstanding technical requirements.

Kind regards,

Muhammad Tarek Abdel Ghafar, M.D

Academic Editor

PLOS ONE

Additional Editor Comments (optional):

Reviewers' comments:

Reviewer's Responses to Questions

**Comments to the Author**

1. If the authors have adequately addressed your comments raised in a previous round of review and you feel that this manuscript is now acceptable for publication, you may indicate that here to bypass the “Comments to the Author” section, enter your conflict of interest statement in the “Confidential to Editor” section, and submit your "Accept" recommendation.

Reviewer #1: All comments have been addressed

Reviewer #2: All comments have been addressed

2. Is the manuscript technically sound, and do the data support the conclusions?

Reviewer #1: Yes

Reviewer #2: Yes

3. Has the statistical analysis been performed appropriately and rigorously? 

Reviewer #1: Yes

Reviewer #2: Yes

4. Have the authors made all data underlying the findings in their manuscript fully available?

Reviewer #1: No

Reviewer #2: Yes

5. Is the manuscript presented in an intelligible fashion and written in standard English?

Reviewer #1: Yes

Reviewer #2: Yes

6. Review Comments to the Author

Reviewer #1: All questions and comments have been addressed. No further feedback or suggestions. I thank the authors for making the time to integrate these comments into their manuscript to better the scientific content.

Reviewer #2: Thanks for giving me the chance to review the above mentioned article entitled "Threshold of Increase in Oxygen Demand to Predict Mechanical Ventilation Use in Novel Coronavirus Disease 2019: A Retrospective Cohort Study Incorporating Restricted Cubic Spline Regression”. The manuscript is improved, my comments have been addressed in the revised version of the article.

7. PLOS authors have the option to publish the peer review history of their article (what does this mean?). If published, this will include your full peer review and any attached files.

Reviewer #1: No

Reviewer #2: No

---

## [Editor Report · Acceptance letter]

6 Jul 2022

PONE-D-22-06778R1 

Threshold of Increase in Oxygen Demand to Predict Mechanical Ventilation Use in Novel Coronavirus Disease 2019: A Retrospective Cohort Study Incorporating Restricted Cubic Spline Regression 

Dear Dr. Yamamoto:

I'm pleased to inform you that your manuscript has been deemed suitable for publication in PLOS ONE. Congratulations! Your manuscript is now with our production department. 

Kind regards, 

on behalf of

Prof Muhammad Tarek Abdel Ghafar 

Academic Editor

PLOS ONE